

# Shallow epifaunal sea cucumber densities and their relationship with the benthic community in the Okinawa Islands

Kohei Hamamoto[1,2], Angelo Poliseno[1], Taha Soliman[1,3] and James Davis Reimer[1,4]

[1] Molecular Invertebrate Systematics and Ecology Lab, Graduate School of Engineering and Science, University of the Ryukyus, Nishihara, Okinawa, Japan
[2] Geological Survey of Japan, National Institute of Advanced Industrial Science and Technology (AIST), Higashi, Tsukuba, Ibaraki, Japan
[3] National Institute of Oceanography and Fisheries, Cairo, Egypt
[4] Tropical Biosphere Research Center, University of the Ryukyus, Nishihara, Okinawa, Japan

## ABSTRACT

Sea cucumbers are important ecological engineers in marine ecosystems. However, the fishery demand of some species, especially large-epifaunal and commercially used (LEC) sea cucumbers, has risen drastically, resulting in serious depletion of local populations for many species. Despite this problem, basic ecological data on sea cucumbers, such as population densities and preferred habitats, are often still insufficient. Here, we report on the population densities of multiple LEC sea cucumber species, and their ambient benthic communities at eight sites around Okinawa Islands. Further, we discuss the correspondence between sea cucumber densities and the surrounding coral communities. Our results show two sites within national or quasi-national parks, Aka and Manza, where stricter rules have been placed on fisheries and land reclamation compared to other areas, had the highest and third highest sea cucumber population densities among sites, respectively. *Holothuria atra* was observed at all survey sites and made up the majority of sea cucumber populations at all sites except for Chatan and Sesoko, where *Holothuria leucospilota* and *Stichopus chloronotus* were most abundant, respectively. Regarding the relationships between benthic composition and LEC sea cucumber species, *S. chloronotus* was significantly correlated with dead corals, scleractinian corals, and coralline algae. As well, *H. leucospilota* had significant correlations with rubble. Although there were no significant correlations between any specific scleractinian coral genus and sea cucumber densities, *S. chloronotus* was marginally insignificant with *Platygyra* and *Psammocora*. Notably, medium- to highly valued species were sparse in our surveys, and most of them appeared at only one site. Additionally, at one site (Odo), only three LEC sea cucumber individuals were observed. Combining these facts with relatively low population densities around the Okinawa Islands compared to densities reported in previous research from the Indo-West Pacific Ocean region, we conclude that Okinawan LEC sea cucumber populations have been and are being impacted by high levels of direct (*e.g.*, overexploitation, as well as coastal development) and indirect anthropogenic pressure (*e.g.*, decreasing water quality). To address the current situation, repeated monitoring and more detailed investigations to reveal the drivers that determine LEC sea cucumber species

Corresponding author
Kohei Hamamoto,
koheihamamoto96@gmail.com

aggregations and population densities are urgently needed, along with more robust management of remaining LEC sea cucumber populations.

## INTRODUCTION

Sea cucumbers are abundant in coral-reef coastline benthic communities (*Bakus, 1973*). Most sea cucumber species in these areas feed on small benthic particulate matter and have a significant role as bioturbators (*Purcell et al., 2016*; *Wolkenhauer et al., 2010*) and are thus ecologically important. *Via* their feeding activity, sea cucumbers enhance the productivity of their surrounding ecosystems (*Uthicke & Klumpp, 1998*; *MacTavish et al., 2012*; *Wolkenhauer et al., 2010*). Furthermore, sea cucumbers are even suggested to be crucial for buffering local ocean acidification by increasing the pH through ammonia discharge (*Uthicke, 2001a*; *Schneider et al., 2011*, *2013*). Finally, sea cucumbers often host many symbiotic species (*Purcell et al., 2016*; *Eeckhaut et al., 2004*; *Hamamoto & Reimer, in press*) and hence contribute to local marine biodiversity.

In the past few decades, sea cucumber fisheries have been under intense pressure all over the world following an increase in their commercial demand (*Conand & Byrne, 1993*; *Purcell et al., 2011*; *Conand, 2018*). Most species targeted for fishery purposes are large-epifaunal and commercially used sea cucumbers (hereafter called "LEC sea cucumbers"). In the Western Central Pacific region, most important LEC species inhabit shallow waters (0–30 m, *Kinch et al., 2008*). As these organisms are easy to catch and their recovery from overharvesting is slow (*Uthicke, Welch & Benzie, 2004*), such heavy exploitation has resulted in severe depletion in many regions of the world (*Purcell et al., 2013*), and hence management of shallow water LEC sea cucumbers is important to preserve coastal populations. However, basic data such as population densities, species assemblages, and substrate preferences are still lacking at the local level for most marine regions, yet are essential to set effective and precise conservation strategies. Surprisingly, the food sources of most species are largely unknown, with few exceptions (*e.g.*, *Uthicke, 1999*; *Mfilinge & Tsuchiya, 2016*; *Yamazaki et al., 2019*; *Gao et al., 2014*, *2022*), and thus the main factors that drive sea cucumber distribution patterns are yet to be determined.

In Okinawa Prefecture, southern Japan, the LEC sea cucumber fishery developed recently and rapidly as in other parts of the world. The sea cucumber fishery in Okinawa is usually done by hand-collection while diving in shallow water, along with collecting other target products (*Okinawa General Bureau, 2017*). The annual harvest reached 225 tons in 2011 with a value of 291 million yen (=ca. 2.5 million USD) in 2012 (*Okinawa General Bureau, 2017*). However, overharvesting caused rapid stock depletion and resulted in regulation of the sea cucumber fishery since 2013. This regulation simply prohibits the

collection of sea cucumbers for non-fishers regardless of species, size, or season (Okinawa Prefecture HP, February 2022 accessed). Studies collecting basic information such as population density have been conducted in other locations around the world such as La Reunion (*Conand & Mangion, 2002*), the Philippines (*Dolorosa, 2015*), and Sri Lanka (*Dissanayake & Stefansson, 2012*), but only scant knowledge is available around the Ryukyu Islands (*e.g.*, *Yamana et al., 2020* around Amami-oshima Island; *Tanita & Yamada, 2019* and *Nishihama & Tanita, 2021* around Ishigakijima Island, Ryukyu Islands). Thus, for Okinawa Prefecture and specifically around Okinawajima Island and surrounding islands, there is a need for basic knowledge such as local population densities, species assemblages, and preferred habitats to make more precise and effective management decisions and rules. To address this need, here we conducted population density and benthic composition surveys of shallow water LEC sea cucumbers at multiple sites around Okinawajima Island and some surrounding islands (hereafter the Okinawa Islands). In addition, we surveyed zooxanthellate scleractinian corals at the same sites to investigate potential relationships between holothurian species and coral communities.

# MATERIALS AND METHODS

## Field surveys

Eight locations around the Okinawa Islands were surveyed in this study in the summers of 2020 and 2021; Chatan, Kayou, Kin, Manza, Odo, Sesoko, Uruma and Aka (Table 1, Fig. 1). Each site's characteristics were as follows: Chatan and Odo are typical lagoons near the reef crest; Kayou and Kin are also lagoons but farther from the reef crest than the previous two sites; Manza, Sesoko and Aka are gradual reef slopes without obvious reef edges; Aka has a large artificial wall made of wave-dissipating blocks (tetrapods; *Masucci & Reimer, 2019*); and finally Uruma is a tidal flat without much reef structure. At each location, two different surveys were performed; (i) line intercept transects of benthos, and (ii) 50-min timed swims (Fig. 2). All surveys were done *via* snorkeling, and no sea cucumbers were collected, hence no permits were needed. Detailed methodologies of these surveys are as follows.

(i) Line intercept transects

At each location, six 10 m transect tapes were laid horizontally to the reef crest on the sea bottom, and digital photographs were taken every 50 cm. Time, depth, temperature, and GPS coordinates were recorded at the start and end of each transect. After each survey, photographs were analyzed as detailed below.

(ii) 50-min timed swims

One observer swam 10 min in one direction parallel to shore and took photographs of every LEC sea cucumber individual that appeared with a ruler. Once 10 min had passed, the observer swam a few minutes towards the reef crest (away from shore), and then swam 10 min parallel to shore in the opposite direction compared to the previous 10-min swim. When the reef flat was too narrow (*i.e.*, Sesoko) or the reef became too deep so that a 10-min swim was not possible, the number of replications were reduced (Fig. 1). At the start and end of each 50-min swim, time and GPS coordinates were recorded (Table S1).

**Table 1  General information of survey sites.**

| Location | Site | Area investigated (m²) | Total ind. no. | Total pop. den. (ind/m²) | Water temp. (°C) | Depth (m) |
|---|---|---|---|---|---|---|
| Okinawa Island | Chatan | 840 | 53 | 0.0631 | 29 | 1.0–1.8 |
| | Uruma | 848 | 75 | 0.0884 | 30 | 1.7–1.9 |
| | Kin | 1,352 | 55 | 0.0407 | 29 | 1.0–1.5 |
| | Odo | 1,748 | 3 | 0.0017 | 28–29 | 1.6–2.1 |
| | Kayou | 1,942 | 12 | 0.0062 | 29 | 1.7–2.3 |
| | Sesoko | 1,892 | 14 | 0.0074 | 27 | 1.7–2.5 |
| | Manza | 1,186 | 95 | 0.0801 | 27 | 1.4–1.7 |
| Kerama Islands | Aka | 1,450 | 136 | 0.0938 | 24–25 | 1.2–1.6 |

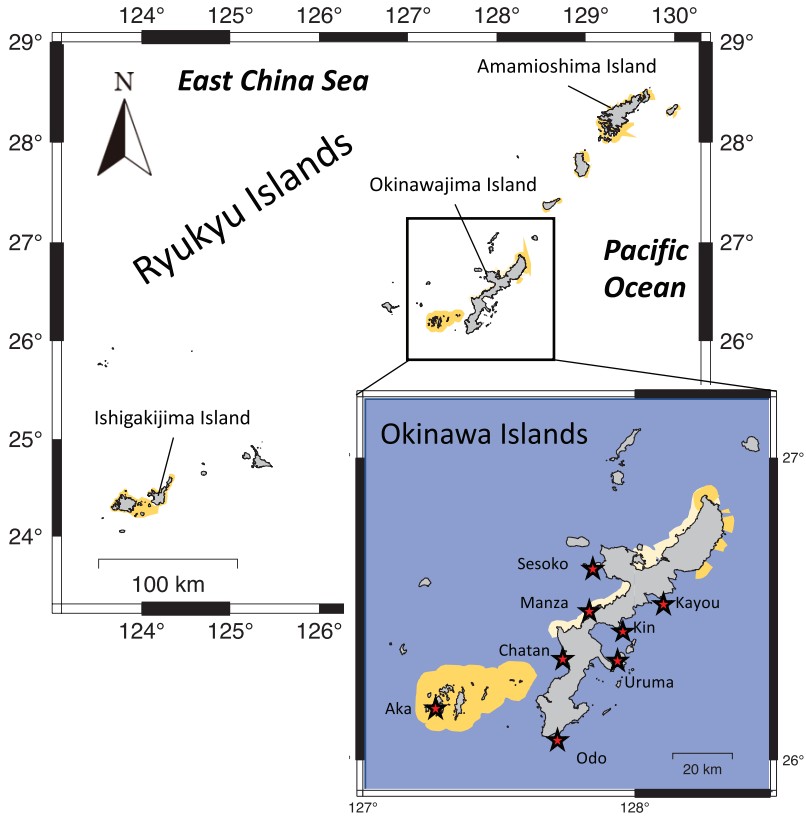

**Figure 1  Map showing the Ryukyu Islands.** Survey sites are shown with red stars in the higher resolution map. Light yellow shows quasi-national park and dark yellow shows national park areas.

## Benthic community investigation and coral identification

For benthic community investigation, benthos was assigned into one of 15 categories; scleractinian corals, dead corals, sponges, *Halimeda* algae, soft corals, coralline algae, macroalgae, turf algae, algal assemblage (multiple types of turf, macro-, and/or coralline algae), seagrass meadow, sand, rubble, rocks, others (=other taxa such sea cucumbers, or abiotic materials such as iron bars, *etc.*), and unidentifiable components (Table S2).

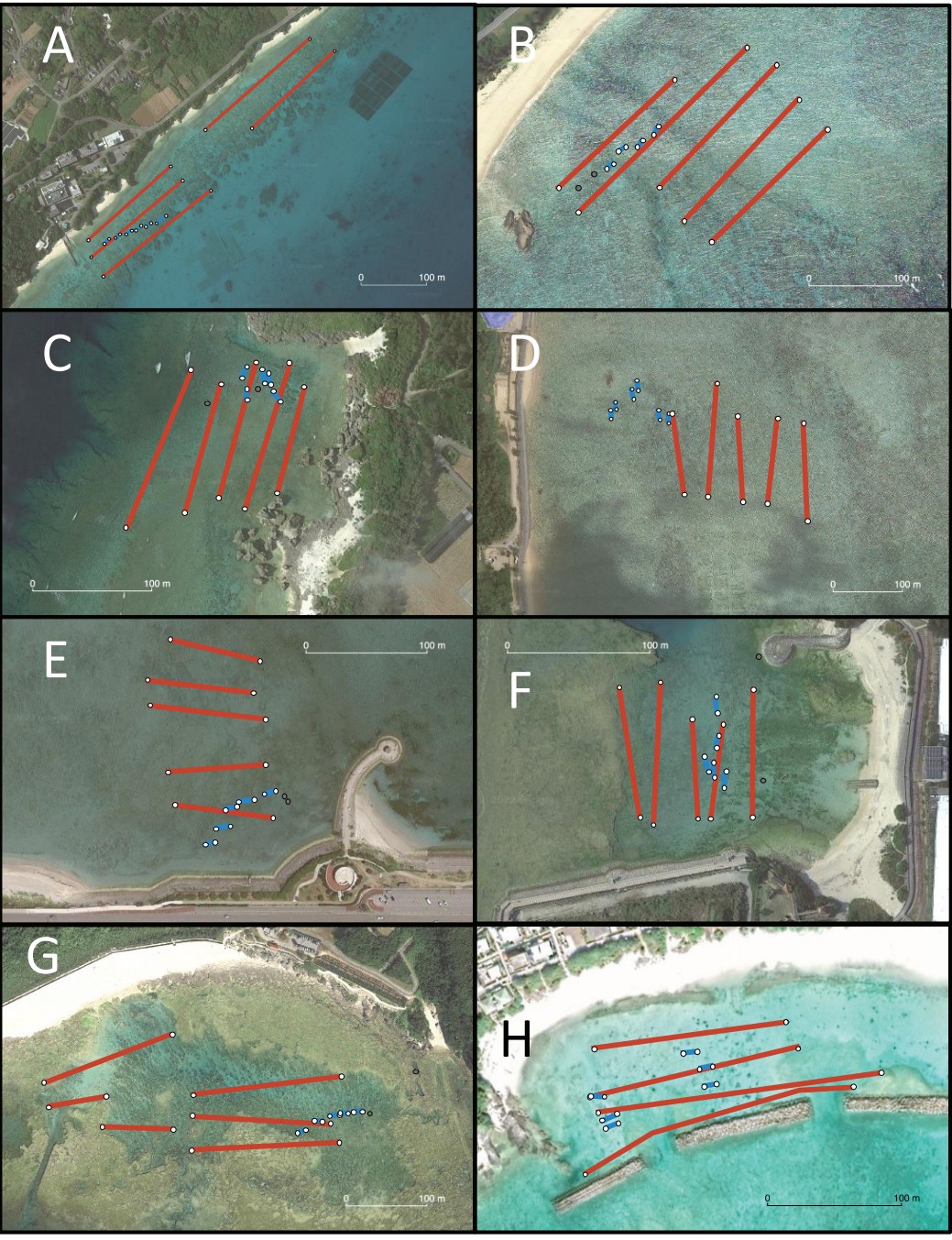

**Figure 2 Aerial photograph of each survey site.** (A) Sesoko, (B) Kayou, (C) Manza, (D) Kin, (E) Uruma, (F) Chatan, (G) Odo, and (H) Aka. Scale is shown in each photograph. Transect track (blue lines) and areas surveyed (red mesh) based on GPS coordinates recorded during the survey are projected in each photograph. Gray points indicate locations calculated from field notes and not GPS. Map data © 2022 Google.                

Using the photographs taken during the survey, identification of scleractinian corals was performed to the genus level following *Nishihira (2019)*.

## Data analyses

All statistical analyses were conducted using R 4.1.2 (*R Core Team, 2021*). Spearman correlation coefficients were calculated using the "chart.Correlation" function in the "PerformanceAnalytics" package (*Peterson & Carl, 2020*) for each LEC sea cucumber species' density, benthic component cover, and coral genera abundance. Subsequently $p$-values were calculated with "cor.test" with a $p = 0.05$ threshold.

To compare benthic components' similarity at each study site, hierarchical clustering was conducted using the "pvclust" package (*Suzuki, Terada & Shimodaira, 2019*). As well, nMDS clustering was carried out to demonstrate the similarity between sites using "metaMDS", "vegdist", "pam", all in the "vegan" package (*Oksanen et al., 2020*). To test statistical significance of the groupings, the "labdsv" package was used (*Roberts, 2019*). The Canonical Correspondence Analysis (CCA) was conducted using "cca" function in the "vegan" package (*Oksanen et al., 2020*), with sea cucumber density as the objective and benthic components as the explanatory variables. For CCA, we only used species that appeared at more than two sites for sea cucumbers and benthic components in characterizing the nMDS grouping.

## RESULTS

Surveys were done around Okinawajima Island from September to October 2020 and in November 2021 at Aka Island. Water temperatures ranged from 27 °C to 30 °C during 2020 surveys and 24 °C to 25 °C for the 2021 survey, and the depth range of surveys was 1.0 to 2.5 m (Table 1). During the survey periods, no strong typhoons occurred in the survey area.

## LEC sea cucumber species, density, and relationship with benthic components

In the course of the surveys, a total of nine LEC sea cucumber species from three families (Synaptidae, Holothuriidae and Stichopodidae) were observed (Table 2; Fig. 3). *Holothuria (Halodeima) atra* Jaeger, 1833 was observed at all survey sites, and was the most abundant species at Kayou, Kin, Manza, Odo, Uruma and Aka. *Holothuria (Mertensiothuria) leucospilota* (Brandt, 1835) and *Stichopus chloronotus* Brandt, 1835 were the most abundant species for Chatan and Sesoko, respectively. *Stichopus chloronotus* was found only at Sesoko, Manza and Aka, where scleractinian corals were one of the main benthic components. However, at Odo, where scleractinian corals were also present, we observed only three individuals of three different sea cucumber species, but no *S. chloronotus*. Some LEC sea cucumber species were observed at only one site with low numbers of individuals (*e.g.*, *Holothuria (Halodeima) edulis* Lesson, 1830 ($n = 1$) and *Actinopyga echinites* (Jaeger, 1833) ($n = 3$) at Chatan, *Holothuria (Metriatyla) scabra* Jaeger, 1833 ($n = 1$) at Uruma, *Bohadschia argus* Jaeger, 1833 ($n = 1$) at Odo and *Synapta maculata* (Chamisso & Eysenhardt, 1821) ($n = 2$) at Aka; Table 2).

**Table 2  Number of individuals of each species observed at survey sites.**

| | Holothuria atra | Holothuria leucospilota | Holothuria edulis | Holothuria whitmaei | Holothuria scabra | Actinopyga echinites | Bohadschia argus | Stichopus chloronotus | Synapta maculata | Total |
|---|---|---|---|---|---|---|---|---|---|---|
| Chatan | 15 (0.0179) | 34 (0.0405) | 1 (0.0012) | – | – | 3 (0.0036) | – | – | – | 0.0631 |
| Uruma | 70 (0.0825) | 4 (0.0047) | – | – | 1 (0.0012) | – | – | – | – | 0.0884 |
| Kin | 55 (0.0407) | – | – | – | – | – | – | – | – | 0.0407 |
| Odo | 1 (0.0006) | – | – | 1 (0.0006) | – | – | 1 (0.0006) | – | – | 0.0017 |
| Kayou | 12 (0.0062) | – | – | – | – | – | – | – | – | 0.0062 |
| Sesoko | 3 (0.0016) | – | – | – | – | – | – | 11 (0.0058) | – | 0.0074 |
| Manza | 48 (0.0405) | – | – | 3 (0.0025) | – | – | – | 44 (0.0371) | – | 0.0801 |
| Aka | 131 (0.0903) | – | – | 1 (0.0007) | – | – | – | 2 (0.0014) | 2 (0.0014) | 0.0938 |

**Note:**
Population density is shown in parentheses (ind/m$^2$).

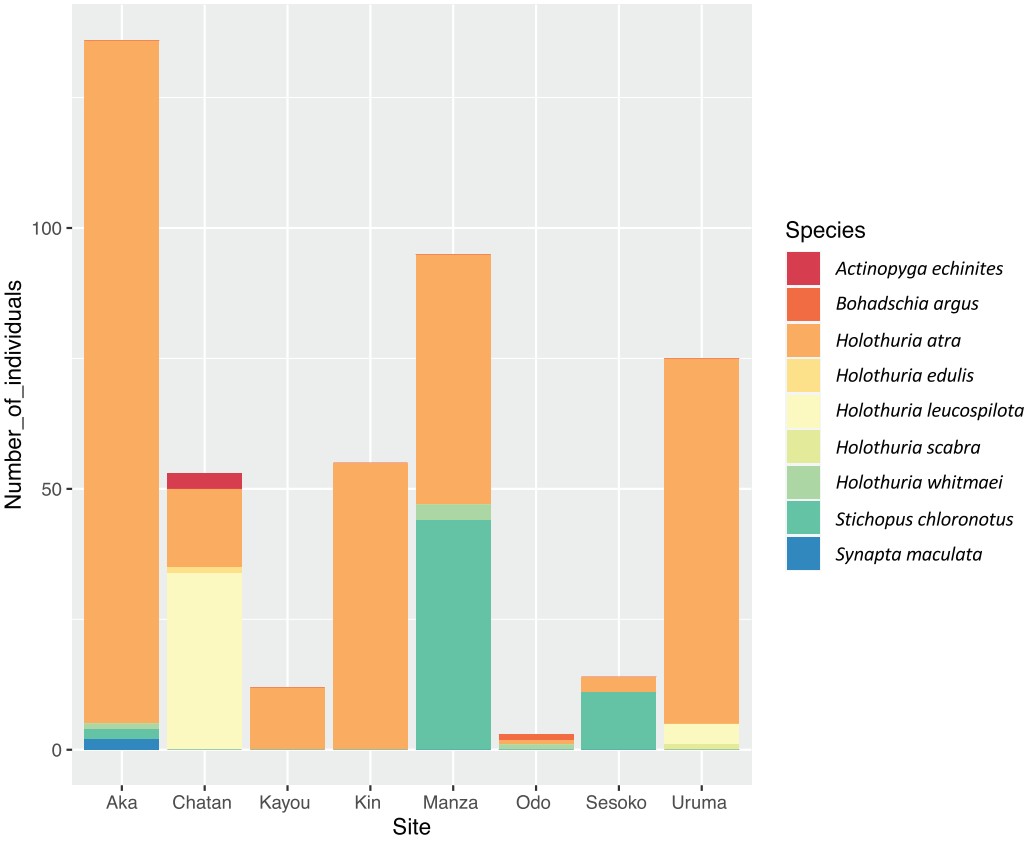

**Figure 3  Number of individuals of sea cucumber species at each site.** Color shows each sea cucumber species, and the y-axis shows the number of individuals.

Total LEC sea cucumber density was highest at Aka Island (=0.0938 ind/m$^2$), followed by Uruma and Manza (=0.0884 and 0.0801 ind/m$^2$, respectively). The lowest density was at Odo (=0.0017 ind/m$^2$). Population densities of species that were observed at more than two survey sites were the highest at Aka for *H. atra*, Chatan for *H. leucospilota*, and Manza for *Holothuria (Microthele) whitmaei* Bell, 1887 and *S. chloronotus*.

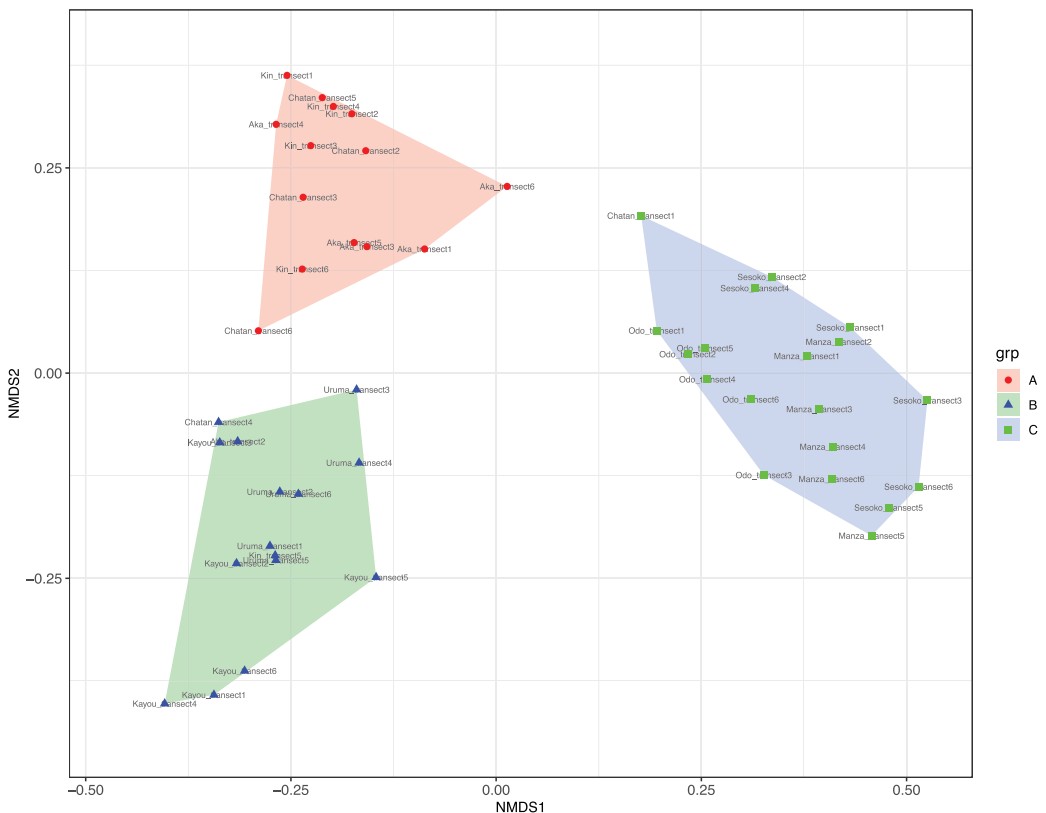

**Figure 4 Result of nMDS analysis.** (A) Sandy bottoms, (B) seagrass meadow with macroalgae, and (C) hard substrate with living scleractinian corals. Colors designate major groups observed.

At Chatan, Kin and Aka, most of the benthic surface was covered by sand, whereas at Kayou and Uruma, more than half of the sea bottom consisted of sea grass meadows. On the other hand, Sesoko, Manza, and Odo had substantial hard substrate coverage with low amounts of scleractinian coral cover (=13% to 19%, Table S2). According to the cluster analysis and nMDS results, the substrates of study locations could be divided into three categories; (i) sandy bottoms, (ii) seagrass meadow with macroalgae, and (iii) hard substrate with living scleractinian corals (Figs. 4 and S1; Table S4). All six transects from the same survey site were assigned into only one nMDS grouping except for Kin, which was assigned into two groups, and Chatan, which was assigned into three groups (Table S5).

Based on the CCA plot, *S. chloronotus* preferred scleractinian corals, dead corals and rocky bottoms that were abundantly seen at Sesoko, Manza and Sesoko (Fig. 5). Macroalgae and seagrass meadows that characterized Kin, Kayou, Uruma and Aka had weak relationships with *H. atra*, whereas *H. leucospilota*, which was abundant at Chatan, did not have any strong relationships with components which characterized major nMDS groupings. Correlation tests between abundant sea cucumber species' (*H. atra*, *H. leucospilota* and *S. chloronotus*) densities and benthic components showed significant correlations between *S. chloronotus* with dead corals ($p$-value = 0.01467), scleractinian corals ($p$-value = 0.01398), and coralline algae ($p$-value = 0.04455, Fig. S2).
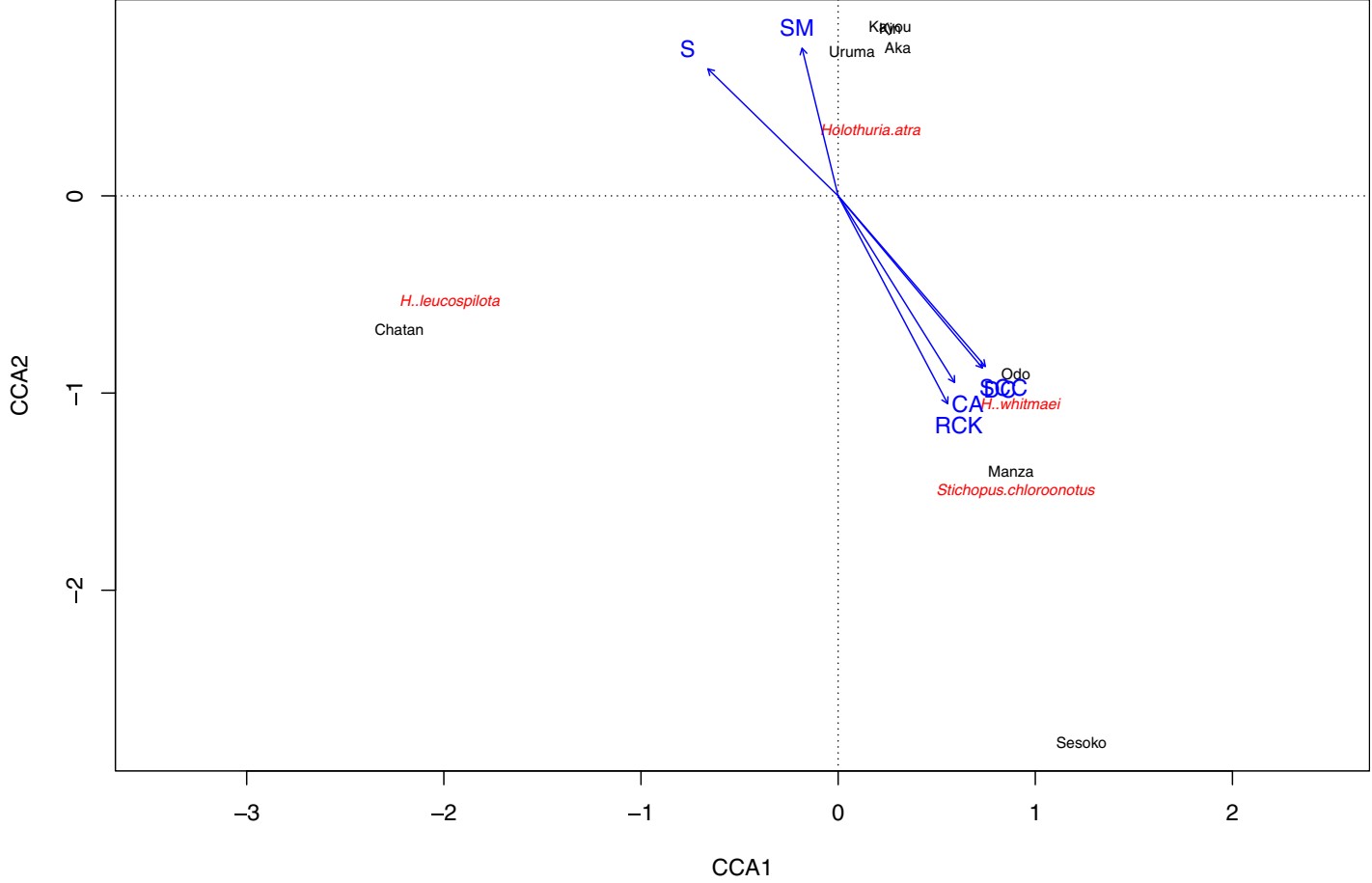

**Figure 5 Result of Canonical correspondence analysis with each sea cucumber species' density as objective variables and sediment and biota types as explanatory variables.** Only sea cucumber species that appeared at more than two sites and benthic components that characterize nMDS grouping were considered as variables.

Also, *H. leucospilota* had significant correlations with rubble and unidentifiable components (*p*-values = 0.03873 and 0.03002, respectively). *H. atra* showed no significant correlation with any benthic component.

In total, 16 scleractinian coral genera were identified in the surveys (Table S3). The most abundant genus was *Montipora* at Manza, Odo and Aka, and *Acropora* at Sesoko. *Montipora* and *Porites* were found at all four sites. The greatest number of scleractinian genera was observed at Sesoko (15 genera) followed by Manza (10 genera), Odo (six genera) and Aka (three genera). None of the comparisons between sea cucumber densities and coral genera had a significant correlation, although *S. chloronotus* with *Platygyra* and *Psammocora* was marginally insignificant (*p*-value = 0.05132, Fig. S3).

## DISCUSSION

### Species composition and population density

Among the survey sites, Chatan and Aka had the highest number of LEC sea cucumber species observed (four species), whereas Kin and Kayou had the lowest (one species).

The high diversity at Aka may be explained by its inclusion in Keramas National Park. On the other hand, it should be noted that Chatan, which is next to a relatively developed coastal area, also had high species diversity. Chatan was the only site in which the transects were assigned into three different nMDS groups, demonstrating that Chatan had a wider variety of benthic components compared to the other surveyed sites. This characteristic may be one cause of the comparatively high diversity of LEC sea cucumber species at this site.

In this study around Okinawajima Island, the population densities of LEC sea cucumber species were generally lower than those observed in previous research conducted in the Indo-Pacific region. For example, *Holothuria atra* at 0.09 ind/m$^2$ was the highest density observed in the current study, while 0.12 to 3.40 ind/m$^2$ were observed at Ishigakijima Island (*Nishihama & Tanita, 2021*), and 0.05 to 0.39 ind/m$^2$ at Mauritius (*Lampe, 2013*) and 10 ind/m$^2$ at New Caledonia (*Purcell, Gossuin & Agudo, 2009*). The low numbers around Okinawajima Island may reflect the overexploitation of LEC sea cucumbers around Okinawajima Island as previously reported (*Okinawa General Bureau, 2017*) or possible habitat degradation due to coastal armoring (*Masucci & Reimer, 2019*; *Masucci, Biondi & Reimer, 2021*). In particular, some medium- to high-value species such as *H. whitmaei*, *H. scabra*, and *Actinopyga echinites* (*Conand & Muthiga, 2007*; *Purcell, Gossuin & Agudo, 2009*; *Purcell, 2010*) showed extremely low densities in the current study (0.0025, 0.0012 and 0.0036 ind/m$^2$ maximums, respectively) even though these species were observed under similar reef conditions in previous studies (*e.g.*, *Purcell, Gossuin & Agudo, 2009*; *Kohler, Gaudron & Conand, 2009*; *Uneputty, Tuapattinaja & Pattikawa, 2017*). Thus, we conclude the low numbers of these species observed in this study likely indicate overharvesting. Furthermore, it has been shown that *S. chloronotus* is also overharvested in East Africa and the Pacific (*Pirog et al., 2019*). As well, based on extremely low mitochondrial genetic diversity, it has been suggested this species has experienced high anthropogenic pressure around Okinawajima Island (*Soliman et al., 2016*). In the present study, the low population density of *S. chloronotus* compared to previous studies (0.04 ind/m$^2$ maximum in current study; 0.09 ind/m$^2$ at Mauritius (*Lampe, 2013*); 1.5–3.6 ind/m$^2$ at La Réunion (*Kohler, Gaudron & Conand, 2009*)) may also show such impacts. However, the number of individuals of *S. chloronotus* observed in our study is similar to those previously reported from Ishigakijima Island (0.04–0.34 ind/m$^2$, *Tanita & Yamada, 2019*), and hence longer-term monitoring of this species is recommended to better ascertain the status of its populations.

Among sites, Odo had the lowest number of individuals (three individuals) and population density (0.0017 total ind/m$^2$). However, it was reported that 1.3 tons of sea cucumbers were harvested in 2014 from the coasts of Itoman City, which includes the Odo coastal area (*Okinawa General Bureau, 2017*), and therefore we speculate that there were substantial amounts of LEC sea cucumbers present at least in the very recent past around Odo. The gap observed between past reports and the current study may be due to slow population recovery after over-exploitation, which has been reported in previous research (*Uthicke, Welch & Benzie, 2004*; *Friedman et al., 2011*; *Rehm et al., 2014*; *Ramírez-González et al., 2020*). The majority of sea cucumbers are gonochoric broadcast spawners (*Babcock*

*et al., 1992*; *Mercier & Hamel, 2009*) with low mobility, and thus considered vulnerable to drastic population size declines due to the "allee effect" (*Purcell et al., 2011*). In other words, once a population size becomes too low, the population can no longer maintain itself and recruitment from elsewhere is needed. However, it has been suggested that there may be no sexual recruitment in sea cucumbers as juveniles are typically not observed in coastal reef areas (*e.g.*, *H. atra* at Taiwan, *Chao, Chen & Alexander, 1994*; *H.atra* and *S. chloronotus* at Great Barrier Reef, *Uthicke, 2001b*; *H. nobilis* at Great Barrier Reef, *Uthicke & Benzie, 2002*), therefore local population recovery may take much longer than expected (*Uthicke, Welch & Benzie, 2004*; *Ramírez-González et al., 2020*).

*Holothuria atra* was observed at all locations and comprised the majority of individuals at five out of eight locations. Similar patterns have been reported in other studies across the Indo-West Pacific Ocean (*Purcell, Gossuin & Agudo, 2009*; *Lampe, 2013*; *Setyastuti, 2015*; *Tanita & Yamada, 2019*). However, it should be noted that the relatively high population density of *H. atra* reported in current study, and perhaps in some previous studies, might reflect its low economic value compared to other more highly-valued species. Among sites, Aka showed the highest number of individuals (131 individuals) and density (0.0903 ind/m$^2$) of *H. atra*, and this site is within Keramashoto National Park (see Fig. 1 for its location), which was reported recently to harbor the highest genetic diversity of this species (*Hamamoto et al., 2021*). As well, Manza, which showed the second highest COI genetic diversity (*Hamamoto et al., 2021*) and had the second highest total individual numbers following Aka, and third highest total population density among sites in the current study, is within a quasi-national park. Combined, these facts may indicate this species is comparatively better conserved within parks. In Japanese national parks, although fisheries are not completely prohibited, stricter rules are in place for the capture of some organisms such as fish, anemones, and scleractinian coral species (*Japanese Ministry of Environment, 1957*). Also, the rules on coastal landfilling (land reclamation), sewage discharge, and construction are stricter (*Japanese Ministry of Environment, 1957*), thus all of these local stressors that potentially impact local populations of benthic organisms such as sea cucumbers are suppressed within those parks. Such higher population densities of holothurian species in protected areas compared to unprotected areas have been reported previously (*Uthicke & Benzie, 2001*; *Rehm et al., 2014*; *Asha et al., 2015*).

## Benthic composition and sea cucumbers' microbiota

Benthic components were divided into three major types: sandy substrate, seagrass meadow with macroalgae, and hard substrate with living scleractinian corals (Fig. 4). The sites characterized by sandy and seagrass substrates were mostly occupied by *H. atra* both in terms of number of individuals and population density, in accordance with previous reports (*Bakus, 1973*; *Purcell, Gossuin & Agudo, 2009*; *Tanita & Yamada, 2019*; *Nishihama & Tanita, 2021*). However, *H. atra* did not have significant correlation with sand or seagrass meadows. On the other hand, *S. chloronotus* showed significant correlations with living scleractinian corals and dead coral cover (*p*-values = 0.01398 and 0.01467, respectively). It has previously been confirmed that *S. chloronotus* has a stronger preference than *H. atra* and *H. edulis* for, and are selectively feeding on, sediments with

higher microalgal organic content (*Uthicke & Karez, 1999*), and therefore the correlation observed here may indicate that there is appropriate organic content around corals and on dead coral rubble. However, no pairs of any coral genus and abundant LEC sea cucumber species had significant correlations, and thus this hypothesis remains to be confirmed. According to our results, only *S. chloronotus* with *Platygyra* and *Psammocora* were marginally insignificant, but as we observed *S. chloronotus* at only three sites, we believe that future studies including additional sites and data such as organic content, other abiotic indices measurement, and microbe metabarcoding may provide further insights to better understand these relationships.

It is widely known that sea cucumber intestinal biota shares components with ambient sediment (*Yamazaki et al., 2019*; *Gao et al., 2022*). Moreover, *Cleary et al. (2019)* found that sea cucumbers, algae, and stony corals' bacterial communities had similar taxonomic composition and diversity, and members of the phylum Planctomycetes were usually abundant. Planctomycetes are commonly found on healthy individuals of the stony coral *Acropora cytherea*, but are scarce in coral colonies affected by skeletal growth anomalies (SGA) (*Rajasabapathy et al., 2020*). As sea cucumbers discharge distinctive microbes that have metabolic capabilities specific to host food digestion (*Yamazaki et al., 2019*; *Jaengkhao, Maeroh & Wanna, 2022*), their intestinal biota is thought to influence the surrounding bacterial community, both in the water column and sediment (*Enomoto, Nakagawa & Sawabe, 2012*; *Bogatyrenko & Buzoleva, 2016*; *Yamazaki et al., 2019*). Therefore, given the importance of sea cucumbers in enhancing ambient productivity and in buffering ocean acidification *via* their feeding activity, the interaction between these organisms with the ambient microbial community, and their ecological roles, need to be further explored.

## Conservation recommendations and conclusions

In this study, relatively low population densities of LEC sea cucumbers were observed around Okinawajima Island and we assume this is likely caused by persistent anthropic impacts such as overexploitation and/or coastal reclamation. In particular, the Odo site showed extremely low densities, possibly due to past or present high fishing pressure, and slow recovery after depletion (*Uthicke, Welch & Benzie, 2004*; *Friedman et al., 2011*). On the other hand, Aka and Manza, both in national or quasi-national parks, had higher population densities, and therefore protection might have successfully affected local benthic conservation. Currently, in Okinawa Prefecture, there are regulations on the harvesting of all sea cucumber species, and therefore capture of any sea cucumber is prohibited for non-licensed citizens. However, there are no rules on fishing activities such as minimum sizes and/or seasonal management. In addition, the sea cucumber fishery does not require expensive equipment as they are slow sedentary organisms. Due to these conditions, sea cucumbers are often sold without fishery association supervision, a situation called "*hamauri*" (="beach-selling") in Okinawa Prefecture. This makes sea cucumber management more difficult, as setting harvest restriction amounts is nearly impossible, and there are no data on what species are actually harvested and from which location. Therefore, the establishment of additional regulations such as requiring the trade

of sea cucumbers to be conducted *via* fishing associations would allow the establishment of more effective additional restrictions. Even with such measures, however, basic reproductive information including spawning season and maturation sizes are yet unknown for most tropical LEC sea cucumbers. Therefore, future ecological investigations such as repeated monitoring at both high-density and low-density sites and deeper water (<30 m) habitat surveys to confirm if Okinawan population are in accordance with the patterns found in other parts of the Ryukyu Islands (*e.g.*, Amamioshima Island, *Yamana et al., 2020*) will provide important knowledge to help develop protection strategies for LEC sea cucumbers in Okinawa Prefecture. As well, to estimate the factor(s) driving each sea cucumber species' density, DNA metabarcoding of bacteria or microalgae in ambient sediments may also be useful.

## ACKNOWLEDGEMENTS

Authors are grateful to Dr. Akira Iguchi (National Institute of Advanced Industrial Science and Technology) for coral identification and statistical advice, and Dr. Gustav Pauly (Florida Museum of Natural History) for sea cucumber identification help. Ms. Marilyn Carletti, Mr. Yusuke Iwaki and Ms. Kairi Takahashi (all from University of the Ryukyus) helped with the field surveys. Dr. Giovanni Diego Masucci (Okinawa Institute of Science and Technology) helped with statistical analyses. We thank two reviewers for their helpful comments on an earlier version of this manuscript.

### Funding

The authors received no funding for this work.

### Competing Interests

James D. Reimer is an Academic Editor for PeerJ.

### Author Contributions

- Kohei Hamamoto conceived and designed the experiments, performed the experiments, analyzed the data, prepared figures and/or tables, authored or reviewed drafts of the article, and approved the final draft.
- Angelo Poliseno conceived and designed the experiments, performed the experiments, prepared figures and/or tables, authored or reviewed drafts of the article, and approved the final draft.
- Taha Soliman analyzed the data, authored or reviewed drafts of the article, and approved the final draft.
- James Davis Reimer conceived and designed the experiments, analyzed the data, authored or reviewed drafts of the article, and approved the final draft.

### Data Availability

  The raw data are available in the Supplemental Tables.

## Supplemental Information

Supplemental information for this article can be found online at http://dx.doi.org/10.7717/peerj.14181#supplemental-information.

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
