# Peer review of "Shallow epifaunal sea cucumber densities and their relationship with the benthic community in the Okinawa Islands"

_PeerJ, doi:10.7717/peerj.14181_

## Round 0.1 · original submission · Minor Revisions

Two expert reviewers have evaluated your manuscript and their comments can be seen below. As you will see the suggestions are important ones that I am sure will improve the manuscript.

·

Basic reporting

This manuscript #75295 must be “accept with minor revision”.

My opinion for rewriting is as follows:

1. In the manuscript, the word “Sea cucumbers” is frequently used, but in this word, authors are using two different meaning, namely Whole the “sea cucumbers”, and the subject species of this manuscript, namely “Large-epifaunal and comarcially used sea cucumbers.” Then, it must be rewrote the latter to another strict word.
2. In the manuscript, Okinawa as an Island, and Okinawa as an administrative organization are mixed, then some readers who has no knowledge for Japanese custom will be confusion. So, it must be rewrote to Okinawajima Island, Ryukyu Islands, as an Island of Okinawa, otherwise it must be rewrote to Okinawa prefecture or Okinawa local government as an administrative organization. Also, Ishigakijima Island must be rewrote to Ishigakijima Island, Ryukyu Islands. Yaeyama Islands must be ignored in this paper, because Ishigakijima Island is regarding only in the reference. Additionally, I hope to authors modify Fig. 1 for that involved more wide area of Southern Japan, in which, Ishigakijima Island, Ryukyu Islands and Amamiohshima Island, Amami Islands, and south part of the Kyushu mainland covered.

Experimental design

3. The explaining is insufficient for the fact that St. Aka had the highest diversity and biomass among the present eight localities, and St. Aka is included into the Kerama National Nature Park. It must be added more information about the Japanese National Nature Park.

4. The explaining is insufficient for the introduction and method, why authors aimed to clarify the distribution of these species only in the shallow less than 2.5 m deep, while these species are mostly distributing in the area under 2.5 m deep.

Validity of the findings

5. The Figure 2 is made by great effort and be excellently detailed, but unfortunately this figure is short of discussion in the manuscript. If author have an idea, I hope to read the discussion for Fig. 2, addition to this, I hope to authors tune-up the contrast of this figure, to several readers, so-called as color-blind.

Additional comments

6. As authors noted, the micro-scale environment are considerable as a reason of distribution patterns of shallow sea cucumbers from the result of nMDS, therefore, what the authors will scope to next? I hope to read the future study plan in the last of this paper.

·

Basic reporting

Well written, Literature adequate and good arrangement of figures and structure of the manuscript

Experimental design

Adequate. Methodology used relevant and able to generate data to answer the objectives.
Although the distribution and substrate affinities of these sea cucumber studied are widely known, there are still gaps in elucidating the mechanism as well as the reasons for such distribution.
This study did not able to answer this question (such as for other studies) as this is done in field observation. However, the densities gave a good indication of the population around Okinawa is valuable

Validity of the findings

Line 212: This is oxymoronic as it is clearly the authors have established that these are high to medium valued sea cucumbers thus prone to overharvesting. Suggest to delete this statement on habitat suitability as they are already suitable as mentioned at the second part of the statement.

Suggestions: the author should elucidate carefully which species were harvested around Okinawa as oppose to consider all species were harvested equally to come with the reason of low density or diversity. The fishing pressure might be targeted more on the high to medium valued materials hence your reading and reasoning might be skewed if they are consider all the same.

Line 243 onwards: As suggested in the last comment, is convenient to just say it is protected area hence higher density. Might be true for the specific area but looking at H. atra wasn't the top priority for fishing. Maybe adding that note would be justified.

Line 264 onwards: S. chloronotus is commonly found on rubbles or even on corals as reported. But the hyphothesis on feeding "... chloronotus feeds on the nutrients produced by corals directly or indirectly through dead coral surfaces.." might not justify this. Clues of the feeding can be found in Uthicke and Karez (1999) or Moriaty (1982). I would take care of making such hyphothesis.

---

## Round 0.2 · Minor Revisions

I have evaluated your manuscript and am satisfied with the changes that you have made in this version. While I was reviewing the manuscript I made a number of suggestions to improve readability in the attached PDF. Please consider them and if you agree, make the changes and upload the new version of the manuscript.

---

## Round 0.3 · accepted · Accept

I am satisfied with the changes made to the manuscript and consider that this manuscript is ready for publication.